# Insights into the Role of Natural Products in the Control of the Honey Bee Gut Parasite (*Nosema* spp.)

**DOI:** 10.3390/ani12213062

**Published:** 2022-11-07

**Authors:** Hesham R. El-Seedi, Aida A. Abd El-Wahed, Yahya Al Naggar, Aamer Saeed, Jianbo Xiao, Hammad Ullah, Syed G. Musharraf, Mohammad H. Boskabady, Wei Cao, Zhiming Guo, Maria Daglia, Abeer El Wakil, Kai Wang, Shaden A. M. Khalifa

**Affiliations:** 1International Research Center for Food Nutrition and Safety, Jiangsu University, Zhenjiang 212013, China; 2Pharmacognosy Group, Department of Pharmaceutical Biosciences, Biomedical Centre, Uppsala University, P.O. Box 591, SE-751 24 Uppsala, Sweden; 3International Joint Research Laboratory of Intelligent Agriculture and Agri-Products Processing (Jiangsu University), Jiangsu Education Department, Nanjing 210024, China; 4Department of Chemistry, Faculty of Science, Menoufia University, Shebin El-Koom 32512, Egypt; 5Department of Bee Research, Plant Protection Research Institute, Agricultural Research Centre, Giza 12627, Egypt; 6Zoology Department, Faculty of Science, Tanta University, Tanta 31527, Egypt; 7Department of Chemistry, Quaid-i-Azam University, Islamabad 45320, Pakistan; 8Department of Analytical Chemistry and Food Science, University of Vigo, 36310 Vigo, Spain; 9Department of Pharmacy, University of Napoli Federico II, 80131 Naples, Italy; 10H.E.J. Research Institute of Chemistry, International Center for Chemical and Biological Sciences, University of Karachi, Karachi 75270, Pakistan; 11Applied Biomedical Research Center, Mashhad University of Medical Sciences, Mashhad 13131-99137, Iran; 12Department of Physiology, Faculty of Medicine, Mashhad University of Medical Sciences, Mashhad 13131-99137, Iran; 13College of Food Science and Technology, Northwest University, Xi’an 710069, China; 14Bee Product Research Center of Shaanxi Province, Xi’an 710065, China; 15School of Food and Biological Engineering, Jiangsu University, Zhenjiang 212013, China; 16Department of Biological and Geological Sciences, Faculty of Education, Alexandria University, Alexandria 215260, Egypt; 17Institute of Apicultural Research, Chinese Academy of Agricultural Sciences, Beijing 100093, China; 18Department of Molecular Biosciences, The Wenner-Gren Institute, Stockholm University, SE-106 91 Stockholm, Sweden

**Keywords:** honey bees, nosemosis, essential oils, plant extracts, active compounds, safety

## Abstract

**Simple Summary:**

The most significant pollinators of crops globally are thought to be honey bees. Unfortunately, bee loss is an issue brought on by a variety of circumstances, such as pesticide use, poor nutrition, parasitic mites, and climate change. The spore-forming unicellular fungi Nosema apis and N. ceranae cause nosemosis, a serious microsporidian disease of adult European honey bees. The disease has an effect on honeybee productivity and reproduction. Antibiotic fumagillin is still used in some countries for the treatment of *Nosema* sp. infection. However, using fumagillin has adverse effects on human health, as well as on honey bee physiology. Therefore, there are trends to develop non-antibiotic alternatives with already existing therapeutics. The present work attempts to emphasize the natural compounds now available for treating nosemosis.

**Abstract:**

The honey bee is an important economic insect due to its role in pollinating many agricultural plants. Unfortunately, bees are susceptible to many pathogens, including pests, parasites, bacteria, and viruses, most of which exert a destructive impact on thousands of colonies. The occurrence of resistance to the therapeutic substances used against these organisms is rising, and the residue from these chemicals may accumulate in honey bee products, subsequently affecting the human health. There is current advice to avoid the use of antibiotics, antifungals, antivirals, and other drugs in bees, and therefore, it is necessary to develop alternative strategies for the treatment of bee diseases. In this context, the impact of nosema diseases (nosemosis) on bee health and the negative insults of existing drugs are discussed. Moreover, attempts to combat nosema through the use of alternative compounds, including essential oils, plant extracts, and microbes in vitro and in vivo, are documented.

## 1. Introduction

Honey bees (*Apis mellifera* L.) are a highly valuable natural resource, and are of great importance to the human population and the whole ecosystem. They play a critical role as pollinators of agricultural crops, wild flora, and natural vegetation, and are an effective biological monitor of environmental contaminants, acting as collectors for airborne particulates and dust deposited on the surfaces on which they land. Honey bees are also source of honey, royal jelly, propolis, bee pollen, bee bread, venom, and wax [1,2,3,4,5,6,7,8,9,10].

A study conducted by Klein and their collaborators using data from 200 countries revealed that fruit, vegetable, or seed production of 87 of the leading global food crops relies upon animal pollination, while only 28 crops do not depend upon animal pollination [11]. The economic gains from agricultural pollination by honey bees are estimated to range between $1.6 and $5.7 billion annually [12]. According to the Food and Agriculture Organization of the United Nations (FAO), three factors affect the occurrence of disease in bees. These factors are (i) queen bees’ genetic origins and hygienic behavior; (ii) pathogenic agents, such as parasites, bacteria, viruses, fungi, and protozoa, which differ in their degrees of presence, virulence, and infectious load; and (iii) environmental conditions, including humidity, temperature, and presence of nectar plants, as nectar secondary metabolites reduce the likelihood of parasitic infection, in addition to bad beekeeping practices [13,14]. Many biological pathogens are responsible for bee colony mortality, including parasites such as nosema [15].

Many synthetic products have been registered for treatment, including fumagillin, a chemical treatment originally isolated from *Aspergillus fumigatus* and currently used in synthesized form to combat nosema disease (nosemosis) in apiculture [16,17]. Periodic treatment with fumagillin may lead to numerous drawbacks, including increased risks to human health as its residues may persist in consumed honey, as well as affecting the physiology of honey bees by altering structural and metabolic proteins in their midgut, leading to low concentrations that struggle to effectively suppress nosema disease [16,18,19]. Moreover, sublethal doses of certain miticides can cause bee mortality due to synergetic interactions, such as in the case of applying tau-fluvalinate to bees that have been treated previously with coumaphos [20]. In contrast, natural products are chemical constituents produced by living organisms found in nature, which are mostly safe. For instance, algae, fungi, plants, bacteria, animal, and insect products exhibit diverse pharmacological properties and provide feasible alternative therapeutic benefits, not only for humans but also for animals. Instead of using conventional drugs to overcome honey bee pathogens scientists have increasingly turned to natural resources to minimize the harmful effects of chemical treatments on honey bees and agricultural crops [21,22].

Natural products come in numerous forms, but essential oils, extracts, and active compounds are the most used. Essential oils are mixtures of fragrant and odorless substances that are extracted by steam distillation from raw plant materials. The volatile metabolites produced can vary due to genetics, climate, rainfall, and geographic causes [23]. Essential oils are mainly composed of terpenes (monoterpenes and sesquiterpenes), aromatic compounds (aldehydes, alcohols, phenols, and methoxy derivatives, among others), and terpenoids (isoprenoids) [24]. These constituents are responsible for the various biological effects of essential oils, particularly as repellents and insecticides [25,26].

Our collective focus in this review is to introduce nosema pathogen and how they may be controlled through use of various natural product candidates.

## 2. Nosemosis

Adult honey bees are susceptible to nosemosis (*Nosemosis apium*), which is one of their deadliest and most widespread diseases. It is a fungi-related microsporidian infection caused by the *Nosema* sp., *N. ceranae*, *N. apis*, and *N. neumanni*. *N. ceranae* was first discovered in the Asian honey bee *Apis cerana*, and then in the western honey bee *A. mellifera* [27]. The infection is spread orally when bees feed on contaminated food, pollen, and water. The disease is then spread by spores in the excrement of infected bees [28,29]. Spores of *Nosema* spp. can be found on flowers and transferred to hives with bee pollen. This is a way of nosema spores spreading through the air in an apiary, but it is not applicable in a laboratory. Research has found that the limited space for flying in the cages and the low number of honey bees in the laboratory limited the spread of the spores into the laboratory air [30].

Nosema species can infect any colony member, including workers, drones, and queens, but research on this microsporidium’s pathological effects has primarily focused on workers. Interestingly, *N. ceranae* infection altered primarily the queens’ physiological functions, namely the vitellogenin titer (a measure of fertility and longevity), total antioxidant capacity, and queen mandibular pheromones [31].

Nosemosis causes digestive problems, including pollen digesting problems and diarrhea, with feces staining visible at the hive entrance, as well as provoking languid behavior (feeling of not being entirely awake), a shorter lifespan, and poor foraging. The infection causes a variety of physiological problems, such as a delayed immune response, lipid synthesis, and pheromone and hormone production problems [32]. This results in significant mortality, adult population loss, a reduction in honey production, and potential colony failure. Microspores infect adult bees’ midgut epithelial cells and germinate rapidly in the host cells before infecting adjacent cells (autoinfection). Ingestion of spores during the cleaning process, as well as excrement from infected bees or during feeding, causes reinfection. In addition, recent evidence has suggested that the black queen cell virus (BQCV) is a virus that targets bees that have been infected early with nosema infection and, thus, manifested with damaged midgut epithelial cells [33]. However, the observation of beekeeping colonies across Spain for two years reported contradictory findings, where 87% of the colonies infested with *Nosema* spp. remained viable, with normal honey production and biological development. Therefore, the presence of *Nosema* spp. alone in hives could not count as a risk factor [34]. Infections increased significantly in bees from pesticide-treated colonies compared to bees from control colonies, which explains the indirect effect of pesticides on pathogen load in honey bees [35].

One of the accepted techniques for identifying a nosema infection is microscopy, due to its accuracy [36]. However, this method is insufficient for distinguishing different nosema species because their spore sizes are similar. When compared to microscopic techniques, molecular detection provides reliable, stable, and sensitive quantification. Polymerase chain reaction (PCR) and ultra-rapid real-time quantitative polymerase chain reaction (UR-qPCR) detection have been widely used to diagnose nosema infection [37,38].

Recently, learning approaches and mobile phone-based fluorescence microscopy have been developed for the rapid imaging, detection, and quantification of nosema spores in honey bees [39,40].

Many approaches, such as those grouped under Good Beekeeping Practices and Biosecurity Measures in Beekeeping, are used to prevent infection [41]. Antibiotics, synthetic drugs such as oxytetracycline, fumagillin, organophosphate, formamidine, and pyrethroid acaricides, are commonly used to combat nosemosis as mentioned in Figure 1 [42]. The only approved chemical therapy for the treatment of nosema disease is fumagillin, an antibiotic known for more than 50 years. It has been widely used in apiculture in the USA, however In Europe, the use of fumagillin is strictly forbidden because there is no established maximum residue level [16]. In addition to potential antibiotic residues present in bee products, antibiotic use has the negative effect of destroying gut bacteria, which decreases immune function and increases vulnerability to nosema infection [43]. Natural products provide a number of advantages, including minimal toxicity, safety for hive products, low resistance, and regional availability. Organic acids, phytotherapeutics, essential oils, microorganisms, and polysaccharides are all helpful in the fight against nosema [44].

## 3. Natural Products in Control of Nosema Disease

### 3.1. Essential Oils

A study conducted by Bravo and his collaborators on the essential oils (EOs) of *Cryptocarya alba* indicated the presence of 39 compounds, including three major components, *α*-terpineol, eucalyptol, and monoterpene *β*-phellandrene. A group of infected bees received EO *C. alba* at a range of doses (1, 2, 3, and 4 µg/bee), a further group received fumagillin syrup (240 µg/bee) as a positive control, and infected bees without treatment were included as a negative control group. The results showed that 4 μg EO/bee was the most effective dose, exhibiting an 80% spore inhibition rate, similar to that of fumagillin. On the other hand, EOs had no toxicity on *A. mellifera* (Table 1) [45]. EOs from several plants, such as lemon balm *(Melissa officinalis),* summer savory *(Satureja hortensis),* peppermint (*Mentha piperita*), and coriander (*Coriander sativum*), have demonstrated anti-nosemosis activity and enhanced the longevity of infected honey bees. Six groups of five bees each (experimental modules) were administered the product (Supresor 1, which is a mixture of essential oils derived from melliferous medicinal herbs) at different concentrations of 1 mL, 2 mL, 5 mL, 10 mL, and 50 ml per liter of syrup, with a positive control (infected non-treated), in addition to two negative control groups (uninfected, treated). The result reported no toxicity against bees, even at the elevated concentration of 10 mL (2000 mg etheric oil) per liter of syrup [46], and the optimal dose detected was 5 mL per liter of sugar syrup.

### 3.2. Plant Extracts

The ethanolic extracts of *Artemisia dubia* and *Aster scaber* have anti-nosemosis functions, seen as a reduction in spores of 77%, at a concentration of 100 μg/mL (Table 1) [47]. Aqueous extract of *A. dubia* and *A. scaber* at 1 μg/mL decreased spore levels by 76%. Butanol and ethyl acetate extracts displayed less activity than aqueous extract [48]. Extracts of six adaptogenic plants, including *Ginkgo biloba*, *Panax ginseng*, *Eleutherococcus senticosus*, *Garcinia cambogia*, *Camellia sinensis*, and *Schisandra chinensis,* were tested. The extract of *E. senticosus* root exhibited the strongest effect against nosema [49]. Another study proved the anti-nosemosis activity of aqueous extracts of nest carton produced from a jet-black ant nest (*Lasius fuliginosus*) with no toxicity on healthy bees. Additionally, the extract of birch carton decreased the number of spores by 97.97% [50].

Administration of the ethanolic extract of *Laurus nobilis* L. (Lauraceae) was carried out on infected bees to test its effects against *N. ceranae*. Syrup was enriched with 1% or 10% plant extract, and a control group was fed only on 60% (*w*/*v*) sucrose syrup without spores. The result showed that 10% *L. nobilis* extract inhibited 10% *N. ceranae,* while 1% *L. nobilis* extract was more effective for nosema inhibition after 19 days of treatment than 10%, and caused death of infected bees (Table 1) [51]. When applied at the time of infection or as a preventative measure, the water extract of *Agaricus blazei* mushroom was effective in reducing the number of *N. ceranae* spores without causing any side effects. Regardless of the presence of nosema infection, *A. blazei* increased the expression of the majority of immune-related genes, such as abaecin, hymenoptaecin, defensin, and vitellogenin. Daily food consumption did not differ between the groups, indicating that the extract was well-tolerated and acceptable [52,53]. The defatted seed meals (DSMs) from *Brassica nigra* and *Eruca sativa*, with a known quantity of various glucosinolates, were added for 8 days to feed the infected bees. The concentrations were 2% and 4%, leading to an inhibition of *N. ceranae*, as well as potential nutraceutical benefits, as reflected on the bee lifespan [54].

The effects of methanolic extracts of Chilean native plant leaves ((2%, 4%, 8%, and 16%), *Aristotelia chilensis*, *Ugni molinae*, and *Gevuina avellana*) and propolis (Biobío (BB) and Los Ríos (LR) regions) on *N. ceranae* infection were also investigated, as mentioned in Table 1. It was found that 88% propolis and *U. molinae* extracts are sufficient for the treatment of infection. The survival rate mean values were 6.62, 8.88, and 7.61, respectively, for the two extracts and the infected control [55]. Another study on the effects of ethanolic propolis extract on the longevity and spore load of *N. ceranae*-infected worker bees indicated that propolis caused a significant reduction in nosema spore load compared to control [56]. As shown in Table 1, the highest survival rate was found in the negative control group with 89%, followed by the 70% in propolis group, with values of 86% of the uninfected samples. The 50% propolis group demonstrated a 54% survival rate, whereas the 70% propolis group had 32%, and the 49% ethanol extract group had 27%. The no-treatment group showed 10% survival in the infected samples [57]. When honey bees were given propolis ethanol extract before or after infection, mortality, infectivity, and *N. ceranae* infection rates were significantly lower than those of the positive control [58].

The use of the herbal supplements NOZEMAT HERB^®^ and NOZEMAT HERB PLUS^®^ resulted in a statistically significant reduction of *N. ceranae* spore load [59].

ApiHerbfi and Api-Bioxalfi dietary supplements that were used also as anti-*N. ceranae* therapies. Both therapies decreased the prevalence of infections and resulted in a reduction of the amount of *N. ceranae*, where ApiHerb had a higher impact [60].

### 3.3. Isolated Compounds from Natural Products

The anti-microsporidian activity of sulphated polysaccharides of algae was reported in an in vitro assay, where they decreased parasite load and increased the survival of infected bees (Table 1) [61]. Similarly, phenolic compounds isolated from aqueous extract of *A. dubia* and *A. scaber*, namely chlorogenic acid, 3,4-dicaffeoylquinic acid (3,4-DCQA), 3,5-dicaffeoylquinic acid (3,5-DCQA), 4,5-dicaffeoylquinic acid (4,5-DCQA), and coumarin, were screened for their anti-nosema effect and toxicity to bees via in vitro and in vivo assays. The results showed that coumarin, chlorogenic acid, and 4,5-DCQA have more potent anti-nosema effects and are less toxic to bees than the other two compounds. Chlorogenic acid and coumarin showed outstanding anti-nosema activities, even at the lowest concentration (10 µg/mL) [62]. Another study on ten compounds (oregano oil, thymol, carvacrol, *trans*-cinnamaldehyde, tetrahydrocurcumin, sulforaphane, naringenin, embelin, allyl sulfide, hydroxytyrosol) and (chitosan, poly I:C) was aimed at controlling *N*. *ceranae* infections. The infected bees were fed on (166.7, 100, 100, and 50 µg/mL) of the compounds and fumagillin (PC), respectively, in 50% sugar syrup and 4 μL/mL ethanol or distilled water to determine the best dose response. Sulforaphane had the highest effect on spores, with a reduction of up to 64% (Table 1) [63]. Chitosan and peptidoglycan, both natural compounds, inhibited *N. ceranae* spore multiplication. Additionally, chitosan and peptidoglycan promote foraging for both pollen and non-pollen without impairing hygienic behavior [64].

In nature, porphyrins are a class of heterocyclic macrocycle organic substances. The consumption of a diet high in sucrose-protoporphyrin amide [PP(Asp)2] syrup significantly inhibited the growth of microsporidia, reduced mortality in infected honey bees, and prevented the spread of the microsporidia. Additionally, their exosporium layers, which were noticeably deformed, demonstrated the morphological changes. Given that they significantly reduce spore numbers, porphyrins are promising candidates for treating microsporidiosis, especially nosemosis, in honey bees [65].

*Thymus vulgaris* is the source of the naturally occurring essential oil ingredient thymol (3-hydroxy-*p*-cymene). Thymol improved honey bee health by increasing bee survival, immune-related gene levels, and oxidative stress parameter values, as well as decreasing nosema spore loads. However, when applied to nosema-free bees, thymol caused certain honey bee health problems, such as reduced bee longevity and induction of oxidative stress and, thus, beekeepers must take caution when applying it [66].

A dietary amino acid and vitamin complex has the potential to protect honey bees from the immunosuppression caused by *N. ceranae*. When compared to the control, the supplements significantly lowered the number of nosema spores. It also affected the expression of immune-related genes in honey bees infected with *N. ceranae* [67].

**Table 1 animals-12-03062-t001:** Natural treatments of nosemosis caused by microsporidia *Nosema ceranae* and *N. apis* (*Nosema* spp.).

Natural Products/Active Compound	Experimental Design	Experimental Outcomes	Reference
*Cryptocarya alba* (oil)	The infected bees received *C. alba* oil at different doses (1, 2, 3, and 4 µg/bee), the fumagillin group received fumagillin syrup (240 µg/bee), and a control group of healthy bees were fed 10 μL of a plain sucrose solution (in vivo).	4 μg EO/bee was the most effective development-controlling dose, exhibiting 80% spore inhibition, similar to that of fumagillin.	[45]
*Doellingeria scabra* and *Artemisia dubia* (extract)	Infected bees were fed 25 μg/mL extract mixture in 60% sucrose solution for 10 days. One control group of uninfected bees was fed a simple 50% sucrose solution and the other control was for untreated infected bees (in vivo).	The best spore reduction was 51% compared to the untreated infected bees group control; no spores were detected in the uninfected bees group.	[47]
*Laurus nobilis* (extract)	Administration of infected bees with syrup enriched with 1% ethanolic extract of *L. nobilis*. One control group fed only 60% (*w*/*v*) sucrose syrup without spores and another control treatment was used (in vivo).	1% treatment with extract showed spore counts lower than control treatment (6.6 × 10^4^ vs. 2.2 × 10^6^ spores on average). No spores were detected in the uninfected bee group.	[51]
Los Rı’os propolis and *Ugni molinae* (extract)	Feeding infected bees with (2% and 8%) propolis and *U. molinae* extracts in pollen substitute, supplemented with 60% sucrose for 15 days (curative method) and feeding another group (preventive approach) of uninfected bees the same extracts for 5 days, after which the bees were infected and treated with the same extract doses for 15 days. Two controls were used, the infected control (where infected bees were maintained with only pollen substitute without extracts) and the healthy control (uninfected bees were used) (in vivo).	8% propolis and *U. molinae* caused a higher survival rate of infected bees in the curative method than the preventive method compared to the infected control treated with bee bread pollen. The survival rate mean values were 6.62, 8.88, and 7.61, respectively for the two extracts and infected control.	[55]
*Andrographis paniculata* (extract)	Supplying infected bees with 50% sucrose syrup solution containing (1%, 2%, 3%, 5%, and 7%) extract prepared using a water decoction method. NC group treated with pure 50% sucrose syrup and a blank control of healthy bees was treated with 50% sucrose syrup (in vivo).	No spores were detected in the blank control group, 2.90 × 10^7^ and 4.94 × 10^8^ spores/bee were detected in the NC group at 7 and 13 day post infection (dpi), respectively. The spore counts of 7%, 5%, and 3% *A. paniculata* were reduced to 1.48 × 10^8^, 1.68 × 10^8^, and 1.94 × 10^8^ spores/bee, respectively, compared to the negative control.	[68]
*Laurus nobilis* (extract)	Oral administration of (1 × 10^4^ μg/mL) of *L. nobilis* ethanol extract. Control treatments supplied only with sugar syrup 60% (*w*/*v*) (in vivo).	Significantly inhibited *N. ceranae* development spores (3.02 × 10^6^ vs. 7.16 × 10^6^ spores in control treatment) at day 19 dpi. No significant mortality of the adult honeybees was observed.	[69]
*Olea europaea* (extract)	Daily administration of 10 mg/mL extract to infected bees (in vivo).	The rate of infection was reduced by 71% at 7 days and to 99% at 35 days.	[70]
Propolis (extract)	Infected bees receiving oral administration of 2 μL of one of the following treatments, obtained by mixing a 1:1 ratio of 50% (*w*/*v*) sucrose solution with aqueous healthy bee gut homogenate (control), (control + 35% ethanol), (control + 50% Propolis), (10^5^ nosema spores per bee + control), (10^5^ Nosema spores per bee + 35% ethanol) and (10^5^ nosema spores per bee + 50% propolis extract) (in vivo).	Both propolis extracts and ethanol (solvent control) have a positive effect on the lifespan of *N. ceranae* infected bees. However, only propolis caused a significant reduction in nosema spore load.	[56]
Propolis (extract)	Feeding infected bees 0%, 50%, and 70% propolis extract mixed with 20 mL 50% sucrose solution (*v*/*v*). Negative control (NC) group wasn’t infected with *N. ceranae* and not treated with propolis. Propolis control bees were not infected with nosema but instead were treated with 70% propolis, and the last control group was infected with nosema but treated with 49% ethanol, which was used during the extraction (in vivo).	The highest survival rate was found in the NC group with 89%, followed by the 70% propolis treatment uninfected group with 86%, the 50% propolis treatment infected group with 54%, the 70% propolis treatment infected group with 32%, the 49% ethanol extract infected group with 27%, and no treatment infected group with 10% survival, respectively.	[57]
Propolis (extract)	Feeding infected bees 2 μL of 50% (*w*/*v*) sucrose solution containing 0, 50, and 70% (*v*/*v*) propolis extract in water. NC was only 2 μL of the 50% (*w*/*v*) sucrose solution containing 35% ethanol as food for infected bees (in vivo).	The highest infection ratio was found in the NC group, 93 ± 4%, followed by 91.7% in the 0% propolis group, and the lowest infection ratio was found in honey bees treated with 70% propolis (43.7%).	[71]
Pentadecapeptide BPC 157 (supplement)	Feeding infected bees sugar syrup 0.25 l (1:1 wate r: sugar) supplemented with 0.1 μg/mL BPC 157 and another control group fed only sugar syrup (in vivo).	BPC 157 reduced the number of spores compared to the initial spore count (average reduction of 40.3% on 20th day and 68.1% on 30th day) and control honey bee colonies showed lower average reduction of *N. ceranae* spores.	[72]
HO21-F (Formulation based on *Olea europaea* plant extract)	In a laboratory setting, 0.5 g/L HO21-F was administered to bees with *N. ceranae* infection. In the field experiment, infected colonies were treated with HO21-F on days 0, 14, and 21 of the experiment, compared to the control group.	With concentrations of 0.5 g/L, the infection levels were reduced by 83.6% in a laboratory environment without affecting the survival rate. In comparison to the control group, in field conditions, an 88% reduction in infection level at a concentration of 2.5 g/L was observed. By modulating antimicrobial peptide synthesis, the formulation may inhibit *N. ceranae* spore germination and improve honey bee humoral immunity.	[73]
Nozevit^®^ (supplement)	Treating infected bees with 1 mL Nozevit^®^ in 500 mL of a sugar syrup vehicle (4 doses a week). Untreated control colonies received 4 applications of 500 mL sugar syrup alone each week (in vivo).	The treated colonies had a significantly larger adult bee population (7.2 bee combs, mean) than the untreated colonies (3.8bee combs, mean.	[74]
HiveAlive™ (supplement)	Administration of supplement in sugar syrup for two years at a dose of 2.5 mL per liter of syrup, and a positive control (PC) group fed 2.5 g fumidil per liter of syrup. In some seasons, bees were fed only syrup or candy (in vivo).	Reductions in nosema spore counts for the second year were as follows: the control group had a reduction of 10%, while the supplement-treated group had a 51% spore reduction over the same period.	[75]
*Pediococcus acidilactici* (Gram-positive bacteria)	To reach the desired concentration of 10^4^ CFU/mL, the bacteria were supplied in sucrose syrup (1:1). Sucrose syrup supplemented with *P. Acidilactici* was fed to honey bees every 48 h. Daily assessments were made of both the mortality and sucrose intake. Fumagillin (1 g/mL) was used as a positive control.	The infected group consumed significantly more sucrose than the control group. Compared to the control group, the infected group’s survival rate increased significantly after receiving bacterial treatment. The bacterial communities in the gut do not change as a result of the bacteria.	[73]
*Bifidobacteria* and *lactobacilli* (microorganism)	There were four bee groups (bees fed with sugar syrup, bees fed with sugar syrup containing the two microorganisms, bees infected with *N. ceranae* spores and fed with sugar syrup, bees infected *with N. ceranae* and fed with sugar syrup containing the two microorganisms). The *bifidobacteria* and *lactobacilli* concentration was 10^6^–10^7^ cfu/mL of sugar syrup (100 mL of a 1:1 sugar/water) (in vivo).	*N. ceranae* detection as follows: both in (bees infected with *N. ceranae* spores and fed with sugar syrup) and (bees infected with *N. ceranae* and fed with sugar syrup containing microorganisms) groups, 20.7% of samples presented spore counts. The (bees fed with sugar syrup containing microorganisms) group showed the absence of *N. ceranae* in 53.3% of insects, whereas in the (bees fed with sugar syrup group), the detection was positive in all samples.	[76]
*Lactobacillus johnsonii* CRL1647 (metabolites)	Oral administration of 50 mL of bacterial metabolite + 200 mL of syrup 2:1 (sugar:water). One control group received 50 mL of culture sterile media + 200 mL of syrup 2:1, another control group received 250 mL of syrup 2:1 (in vivo).	Adult bee populations at the end of the test increased in the metabolite administration group by 39.5%, 26.0% in the culture media group, and by 12.0% in syrup administration group.	[77]
Cruciferous vegetables/sulforaphane	Infected bees fed (166.7, 100, 100, and 50 µg/mL) of the compounds and fumagillin, respectively, all in 50% sugar syrup and 4 μL/mL ethanol or distilled water helping the compounds to dissolve. Another control involved feeding bees only 5 μL of sugar syrup without spores. Subsequently, different concentrations of the compounds were used with the same procedure to determine the best dose response (in vivo).	At 1.25 mg/mL, spore reduction reached 100% compared to control group.	[63]
Citrus fruit/Naringenin		At 2.08 mg/mL, spore reduction reached 64% compared to control group.	
*Origanum vulgare* (oil)/carvacrol		At 0.10 mg/mL, spore reduction reached 57% compared to control group.	
Thymol and resveratrol (compound)	Feeding infected bees with candies containing 0.12 mg/g thymol and 0.01 mg/g resveratrol. Bee candy was prepared by mixing powdered sugar (85%), sterilized honey (10%), and water (5%). The control was candy containing ethanol (in vivo).	Infection levels in control bees was 230 million spores/bee. In resveratrol-fed bees, the infection level at 25 days was 54 million spores/bee, and bees fed with thymol candy showed a decrease in infection levels at 25 days (68% less compared to 19 day).	[78]
Thymol and resveratrol (compound)	Administration of 0.1 mg/g of thymol in syrup (50% *w*/*v* sucrose solution) and 0.01 mg/g of resveratrol. 3.2 μL/g ethanol was used to aid the compound’s solubility. The control cages consisted of infected bees fed with untreated syrup and candy (in vivo).	Proportion of survivors/spore load showed that bees fed with thymol syrup had the highest value (2.4 × 10^−9^), followed by the group fed with resveratrol syrup (1.9 × 10^−9^), whereas bees fed with control candy gave lower results (1.9 × 10^−10^).	[79]
*Porphyridium marinum* (polysaccharides) (compound)	Infected bees fed a polysaccharide solution at a concentration of 100 μg/mL. PC consisted of an infected group with medium containing fumagillin at 1 μg/mL, and two other groups were used as infected control and uninfected control (in vivo).	The best inhibition of parasite growth was obtained for fumagillin (a reduction of ∼60% the parasite load), followed the polysaccharide treatment, which reduced the parasite load by ∼30%, (40.8 × 10^6^ in the polysaccharide group versus 61.6 × 10^6^ spores/bee in the infected control group.	[61]

## 4. Safety of Natural Products as Treatments of Nosemosis

The natural products are safe when consumed in reasonable quantities. “GRAS” is an acronym for the phrase Generally Recognized as Safe, designated by FDA for natural products [1]. Furthermore, toxicity could be applied to adult workers, larvae, eggs, queens, bees’ enzyme activities, and their energy reserves [2], and to avoid any adverse impact of hosts during the first hour, the mortality must be less than 20% at 72 h [3]. Ariana and his colleagues concluded that a spray of 2% thyme, spearmint, and savory essential oil has no harmful effect on honey bees compared to dillsun essence, which caused 12% honey bee mortality [12]. Another study estimated that an essential oil of *Citrus paradisi* and *C. sinensis* reduced Paenibacillus, and at the tested concentrations of 2.34 mg/L, 2.08 mg/L, 1.82 mg/L, 1.56 mg/L, and 1.30 mg/L, had no mortality of honey bees [15]. Screening the toxicity activity of 13 crude plant extracts, *Piper ribesioides* and *P. sarmentosum* were found to be highly toxic to bees but *Thunbergia laurifolia*, *Allium sativum*, *Cymbopogon citratus*, and *Senna alata* had no toxic effect on adult bees, even those exposed to a high concentration [16]. In addition, the metabolites produced by *Lactobacillus johnsonii,* mainly consisting of lactic acid, did not cause bee mortality after 72 h of exposure by oral administration, even in high doses (60 µL of bacterial metabolite produced 10% mortality) [77]. Thymol is among the main chemical constituents of thymus essential oils, which, when applied to other plants, can help in the reduction of bee diseases. For bees infected with nosema, candies containing 0.12 mg/g thymol can decrease bee mortality and increase bee survival [10]. Administration of syrup enriched with 1% or 10% *L. nobilis* leaf extracts is effective inhibiting in vivo development of nosema in bees, depending on the dosage applied. The concentration of 10% was more effective than 1% and acted in a shorter time frame, but caused higher mortality [7].

## 5. Conclusions

Honey bees are well known as one of the most significant pollinators, contributing greatly to the global food supply by pollinating a wide range of crops. Bees, like other living organisms, suffer from many diseases, including nosema parasites. Instead of utilizing conventional pharmaceuticals to combat honey bee infections, scientists have turned to natural resources to reduce the adverse effects of chemicals on honey bees and agricultural crops, as much as possible. In this review we have presented in vitro and in vivo studies that have been conducted on natural solutions for the control and prevention of bee diseases, based on the use of essential oils, plant extracts, and microbes. We have also addressed the toxicity of common natural products on adult honey bees.

However, further research is required to estimate the toxicity of these products on bees’ enzyme activities and their effects on bee energy reserves. We recommended that researchers continue conduct further studies on the use of natural products and monitor toxicity resulting from their use, as well as seeking out new sources of natural products, such as marine organisms and microorganisms. We also recommend further investigation of potential uses for nanotechnology, on a larger scale, to increase the effectiveness of these products. In addition, pharmacists must pay attention to all natural products tested on bee diseases for applications in targeted drugs.

## Figures and Tables

**Figure 1 animals-12-03062-f001:**
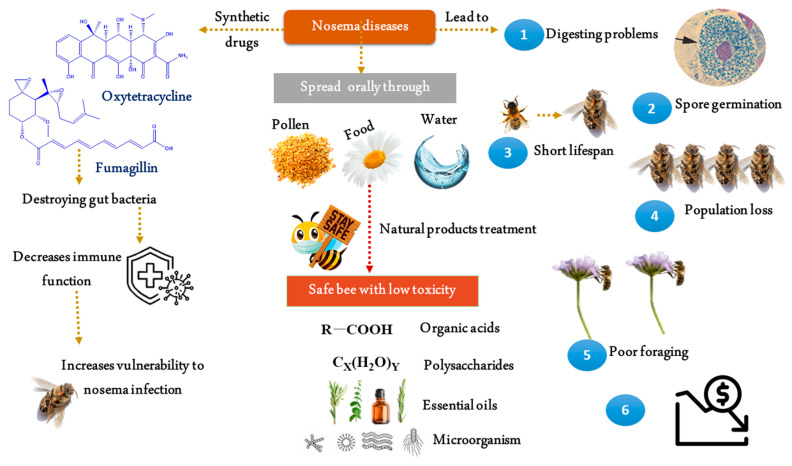
Treatment of nosema diseases in *Apis mellifera* across synthetic and natural products.

## Data Availability

Not applicable.

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
