# Peer review of "Insights into the Role of Natural Products in the Control of the Honey Bee Gut Parasite (Nosema spp.)"

_animals, 2022, doi:10.3390/ani12213062_

Round 1

Reviewer 1 Report

Some minor technical errors are found.

Line 79: Aspergillus fungus

Line 99: sesquiterpenes

Line 126: nosema

Line 210: anti-microsporidian

Line 210: sulfated

Line 213: 3,4-dicaffeoylquinic acid

Line 220: trans-cinnamaldehyde

Line 226: microsporidia Nosema ceranae

Line 232: could be applied to adult workers,

Line 236: maybe it is “dill sun essence

Line 259: as much as possible

Author Response

Some minor technical errors are found.

  • Line 79: Aspergillus fungus

Response: Changed to “Aspergillus fumigatus

  • Line 99: sesquiterpenes

Response: Adjusted

  • Line 126: nosema

Response: Adjusted

  • Line 210: anti-microsporidian

Response: Changed to “anti-microsporidian

  • Line 210: sulfated

Response: “sulphated”

  • Line 213: 3,4-dicaffeoylquinic acid

Response: Adjusted

  • Line 220: trans-cinnamaldehyde

Response: Adjusted

  • Line 226: microsporidia Nosema ceranae

Response: Changed to “microsporidia Nosema ceranae”

  • Line 232: could be applied to adult workers,

Response: Adjusted

  • Line 236: maybe it is “dill sun essence”

Response: Correct

  • Line 259: as much as possible

 Response: Modified

Reviewer 2 Report

The authors attempted to write a review about analysed natural-based products with potential in Nosema sp. control.

General comments and main shortcommings:

(1)   The topic is relevant in the field, but not original, as there are (comprehensive) papers with the same topic published recently (2021 and 2022).

- Chaimanee et al. (2021) https://pubmed.ncbi.nlm.nih.gov/34728218/

- Formato et al. (2022) https://www.mdpi.com/2076-3417/12/2/783

- Iorizzo et al. (2022) https://www.mdpi.com/2309-608X/8/5/424

(2) many omitted investigations closely related to the main aim, e.g.

- Michalczyk et al. (2016) https://sciendo.com/it/article/10.1515/acve-2016-0009

- PtaszyÅ„ska et al. (2018) https://www.nature.com/articles/s41598-018-23678-8

- CiIlia et al. (2020) https://www.mdpi.com/2306-7381/7/3/125

- Valizadeh et al. (2020) https://pubmed.ncbi.nlm.nih.gov/32858847/

- Naree et al. (2021) https://www.tandfonline.com/doi/abs/10.1080/00218839.2021.1905374

- Glavinic et al. (2021b) https://www.mdpi.com/2075-4450/12/10/915

- Nanetti et al. (2021) https://pubmed.ncbi.nlm.nih.gov/33924845/

- Shumkova et al. (2021) https://www.mdpi.com/2076-0817/10/2/234

- Glavinic et al. (2022) https://www.mdpi.com/2075-4450/13/7/574

- Shumkova et al. (2022) http://www.actaveterinaria.rs/volume/issue/22/113/1062

(2) Poor English (many linguistic errors, improper grammar ... ).

Additional comments:                                        

Structure of the manuscript: To my opinion, the Section 2 (‘Nosemosis’) is redundant as it contains only well-known facts. 

Key words: Honeybee should be written ’honey bee’. Other two key words are already present in the title, so I suggest to replace them with: nosemosis, essential oils, plant extracts. 

Honeybee should be written ’honey bee’ also in lines 154 and 263.
Figure 1 is original and interesting, but has typographic error (e.g. nosma).

The title of the Table 1 is not adequate. 

There are also other comments inserted directly in manuscript (attached PDF).

Author Response

The authors attempted to write a review about analysed natural-based products with potential in Nosema sp. control.

General comments and main shortcommings:

(1)   The topic is relevant in the field, but not original, as there are (comprehensive) papers with the same topic published recently (2021 and 2022).

- Chaimanee et al. (2021) https://pubmed.ncbi.nlm.nih.gov/34728218/

- Formato et al. (2022) https://www.mdpi.com/2076-3417/12/2/783

- Iorizzo et al. (2022) https://www.mdpi.com/2309-608X/8/5/424

Response: We would like to thank the reviewer for taking the time and effort necessary to provide such insightful comments. Authors do modifications throughout the manuscript to cover this point.

(2) many omitted investigations closely related to the main aim, e.g.

- Michalczyk et al. (2016) https://sciendo.com/it/article/10.1515/acve-2016-0009

- Glavinic et al. (2017) https://journals.plos.org/plosone/article?id=10.1371/journal.pone.0187726

- PtaszyÅ„ska et al. (2018) https://www.nature.com/articles/s41598-018-23678-8

- CiIlia et al. (2020) https://www.mdpi.com/2306-7381/7/3/125

- Valizadeh et al. (2020) https://pubmed.ncbi.nlm.nih.gov/32858847/

- Naree et al. (2021) https://www.tandfonline.com/doi/abs/10.1080/00218839.2021.1905374

- Glavinic et al. (2021a) https://www.mdpi.com/2075-4450/12/4/282

- Glavinic et al. (2021b) https://www.mdpi.com/2075-4450/12/10/915

- Nanetti et al. (2021) https://pubmed.ncbi.nlm.nih.gov/33924845/

- Shumkova et al. (2021) https://www.mdpi.com/2076-0817/10/2/234

- Glavinic et al. (2022) https://www.mdpi.com/2075-4450/13/7/574

- Shumkova et al. (2022) http://www.actaveterinaria.rs/volume/issue/22/113/1062

 Response: The references are cited throughout the manuscript (lines 233-243, 274-278, and 300-319).

(3) Poor English (many linguistic errors, improper grammar ... ).

Response: The authors have done a second revision and English editing.

(4) Additional comments: 

  • Structure of the manuscript: To my opinion, the Section 2 (‘Nosemosis’) is redundant as it contains only well-known facts. 

Response: This section adjusted and incorporated new data (Lines 136-145, 150-174, and 178- 182).

  • Key words: Honeybee should be written ’honey bee’. Other two key words are already present in the title, so I suggest to replace them with: nosemosis, essential oils, plant extracts. 

Response: Keywords: Honey bees, nosemosis, essential oils, plant extracts, active compounds, safety. 

  • Honeybee should be written ’honey bee’ also in lines 154 and 263.

Response: Adjusted
- Figure 1 is original and interesting, but has typographic error (e.g. nosma).

Response: Adjusted

  • The title of the Table 1 is not adequate. 

Response: Table 1. Natural treatments of nosemosis disease caused by microsporidia Nosema ceranae and Nosema apis (Nosema spp.)”

(5) The response to comments on PDF files

  • Lines 60-64: Please divide the sentence.

Response: Done

  • You did not mention bad beekeeping practice as a main problem. Please consult Stanimirovic et al. (2019) Acta Veterinaria 69(1) 1-31 https://sciendo.com/article/10.2478/acve-2019-0001

Response: Adjusted and the mentioned reference cite

  • Lines 85-85: Please divide the sentence.

Response: Adjusted

  • Line 193: “change 'nosma' to 'Nosema' on picture”

Response: Done

  • Line 194: Natural Product in control of Nosema disease

Response: Adjusted

  • Inadequately written for the review paper (too much details of experiments instead of briefly reported main findings).

Response: Adjusted (Lines 254-256, 270-274).

Line 166: you did not introduce those abbreviations

Response: “Butanol and ethyl acetate”

The title of Table 1. is non-adequate since it does not reflect the content of the Table.

Response: Changed to “Table 1. Natural treatments of nosemosis caused by microsporidia Nosema ceranae and Nosema apis (Nosema spp.)”

Line 227: I suggest to remove 'strategies'.

Response: Done “Safety of natural products as treatments of nosemosis”

Reviewer 3 Report

This review is summarized in a very easy-to-understand way to defend Nosema disease. There are no correction items.

Author Response

This review is summarized in a very easy-to-understand way to defend Nosema disease. There are no correction items.

Response: We would like to thank the reviewer for the nice words

Round 2

Reviewer 2 Report

The authors substantially improved the manuscript. The authors substantially improved the manuscript. I found some minor errors: 

Line 178: In subtitle, one letter "s" is missing (it should be "Natural products in control of Nosema disease"). 

The authors need to follow the rules of writing scientific names:

Line 226. In title of Table 1. instead of Nosema apis (Nosema spp), please write only N. apis. 

Writing only "Nosema" (on many places in your manuscript) is not proper.

(Lines 375-376), reference No. 13  should be corrected, as it is published in journal Acta Veterinaria-Beograd (not in Acta Vet Brno). Please open the link below and look at the upper right corner of original PDF version:   

http://www.actaveterinaria.rs/uploads/documents/6325b1bd5f076a1bf318663535cb0efb30a479eb_Stanimirovic.pdf

Author Response

The authors substantially improved the manuscript. The authors substantially improved the manuscript. I found some minor errors: 

  • Line 178: In subtitle, one letter "s" is missing (it should be "Natural products in control of Nosema disease"). 

     Response: Adjusted

  • The authors need to follow the rules of writing scientific names:

    Response: Adjusted

  • Line 226. In title of Table 1. instead of Nosema apis (Nosema spp), please write only N. apis.

     Response: Done

  • Writing only "Nosema" (on many places in your manuscript) is not proper.

   Response: Adjusted

  • (Lines 375-376), reference No. 13  should be corrected, as it is published in journal Acta Veterinaria-Beograd (not in Acta Vet Brno). Please open the link below and look at the upper right corner of original PDF version:   http://www.actaveterinaria.rs/uploads/documents/6325b1bd5f076a1bf318663535cb0efb30a479eb_Stanimirovic.pdf

  Response: Actually, it was written as an abbreviation in accordance with the Journal's formatting.